# Generalizing Bayesian Optimization with Decision-theoretic Entropies

**Willie Neiswanger**[*], **Lantao Yu**[*], **Shengjia Zhao, Chenlin Meng, Stefano Ermon**
Computer Science Department, Stanford University
Stanford, CA 94305
{neiswanger,lantaoyu,sjzhao,chenlin,ermon}@cs.stanford.edu

## Abstract

Bayesian optimization (BO) is a popular method for efficiently inferring optima of an expensive black-box function via a sequence of queries. Existing information-theoretic BO procedures aim to make queries that most reduce the uncertainty about optima, where the uncertainty is captured by Shannon entropy. However, an optimal measure of uncertainty would, ideally, factor in how we intend to use the inferred quantity in some downstream procedure. In this paper, we instead consider a generalization of Shannon entropy from work in statistical decision theory [13, 39], which contains a broad class of uncertainty measures parameterized by a problem-specific loss function corresponding to a downstream task. We first show that special cases of this entropy lead to popular acquisition functions used in BO procedures such as knowledge gradient, expected improvement, and entropy search. We then show how alternative choices for the loss yield a flexible family of acquisition functions that can be customized for use in novel optimization settings. Additionally, we develop gradient-based methods to efficiently optimize our proposed family of acquisition functions, and demonstrate strong empirical performance on a diverse set of sequential decision making tasks, including variants of top-$k$ optimization, multi-level set estimation, and sequence search[2].

## 1 Introduction

Bayesian optimization (BO) is a popular method for efficient global optimization of an expensive black-box function, which leverages a probabilistic model to judiciously choose a sequence of function queries. In BO, there are a few key paradigms that motivate existing methodologies. One paradigm is decision-theoretic BO, which includes methods such as *knowledge gradient* [16] and *expected improvement* [33, 25]. At each iteration of BO, these methods aim to make a query that maximally increases the expected value, under the posterior, of a final estimate of the optima (sometimes referred to as a *terminal action*). Another common paradigm is based on maximal uncertainty reduction and includes information-based BO methods such as the family of *entropy search* methods [22, 24, 47, 35]. At each iteration of BO, these methods aim to make a query that most reduces the uncertainty, under the posterior, about a quantity of interest (such as the location of the optima).

In the uncertainty-reduction paradigm, the information-based methods have predominantly used Shannon entropy as the measure of uncertainty. While Shannon entropy is one measure of uncertainty that we could aim to reduce at each iteration of BO, it is not the only measure, and it is not necessarily the most ideal measure for every optimization task. For instance, an optimal uncertainty function would, ideally, factor in how we intend to *use* the final uncertainty about an inferred quantity in some downstream procedure.

---

[*]The first two authors contributed equally to this work.

[2]For additional details, see the project website: https://willieneis.github.io/hes-website

36th Conference on Neural Information Processing Systems (NeurIPS 2022).

In this paper, we develop a framework that aims to first unify and then extend these two paradigms. Specifically, we adopt a generalized definition of entropy from past work in Bayesian decision theory [13, 39, 21], which proposes a family of *decision-theoretic entropies* parameterized by a problem-specific loss function and action set. This family includes Shannon entropy as a special case. Using this generalized entropy, we can view information-based BO methods as instances of decision-theoretic BO, with a terminal action chosen from a different type of action set. Similarly, this framework also includes as special cases the decision-theoretic methods such as expected improvement and knowledge gradient, which yields an uncertainty-reduction view of these methods. Beyond this unified view, our framework can be easily adapted to novel problem settings by choosing an appropriate loss and action set tailored to a given downstream use case. This allows for handling new optimization scenarios that have not previously been studied and where no BO procedure currently exists.

As an example, there are many real-world problems where we want to estimate a set of optimal points, rather than a single global optimum. Use cases include when we wish to find a set of highest-value points subject to some constraints on the similarity between these points (e.g. to produce a diverse set of candidates in drug or materials design [38, 44, 45]), or points which satisfy some sequential relation (e.g. to construct a library of molecules that attains a sequence of desired measurements [15]). Further, we may wish to estimate other properties of a black-box function, such as certain curves, surfaces, or subsets of the domain [54, 28, 40]. Due to the vast number of possibilities, most custom problem settings have not been explicitly studied in the literature. A key advantage of our framework is that it provides a way to approach these problems where no suitable methods have been developed.

Additionally, since we define this family of generalized entropies in a standardized way, we can develop a common acquisition optimization procedure, which applies generically to many members of this family (where each member is induced by a specific loss function and action set). In particular, we develop a fully differentiable acquisition optimization method inspired by recent work on one-shot knowledge gradient procedures [4]. This yields an effective and computationally efficient algorithm for many optimization and sequential decision making tasks, as long as the problem-specific loss function is differentiable. In summary, our main contributions are the following:

- We propose an acquisition function based on a family of *decision-theoretic entropies* parameterized by a loss function $\ell$ and action set $\mathcal{A}$. Under certain choices of $\ell$ and $\mathcal{A}$, we can view multiple BO acquisition functions in a single decision-theoretic perspective, which sheds light on the settings for which each is best suited.

- By selecting a suitable $\ell$ and $\mathcal{A}$, we can produce a problem-specific acquisition function, which is tailored to a given downstream use case. This yields a customizable BO method that can be applied to new optimization problems and other sequential decision making tasks, where no applicable methods currently exist.

- We develop an acquisition optimization procedure that applies generically to many instance of our framework. This procedure is computationally efficient, using a gradient-based approach.

- We demonstrate that our method shows strong empirical performance on a diverse set of tasks including top-$k$ optimization with diversity, multi-level set estimation, and sequence search.

## 2  Setup

Let $\mathbf{f} : \mathcal{X} \to \mathcal{Y} \subset \mathbb{R}$ denote an expensive black-box function that maps from an input search space $\mathcal{X}$ to an output space $\mathcal{Y}$, and $\mathbf{f} \in \mathcal{F}$. We assume that we can evaluate $\mathbf{f}$ at an input $x \in \mathcal{X}$, and will observe a noisy function value $y_x = \mathbf{f}(x) + \epsilon$, where $\epsilon \sim \mathcal{N}(0, \eta^2)$.

We also assume that our uncertainty about $\mathbf{f}$ is captured by a probabilistic model with prior distribution $p(f)$, which reflects our prior beliefs about $\mathbf{f}$. Given a dataset of observed function evaluations $\mathcal{D}_t = \{(x_i, y_{x_i})\}_{i=1}^{t-1}$, our model gives a posterior distribution over $\mathcal{F}$, denoted by $p(f|\mathcal{D}_t)$.

Suppose that, after a given BO procedure is complete, we intend to choose a terminal action $a$ from some set of actions $\mathcal{A}$, and then incur a loss based on both this action $a$ and the function $\mathbf{f}$. We denote this loss as $\ell : \mathcal{F} \times \mathcal{A} \to \mathbb{R}$. As one example, after the BO procedure, suppose we make a single guess for the function maximizer, and then incur a loss based on the value of the function at this guess. In this case, the action set is $\mathcal{A} = \mathcal{X}$ and the loss is $\ell(\mathbf{f}, a) = -\mathbf{f}(a)$.

# 3 Decision-theoretic Entropy Search

In this section, we first describe a family of *decision-theoretic entropies* from past work in Bayesian decision theory [13, 39, 21], which are parameterized by a problem-specific action set $\mathcal{A}$ and loss function $\ell$. This family includes Shannon entropy as a special case. We denote this family using the symbol $H_{\ell,\mathcal{A}}$, and refer to it as the $H_{\ell,\mathcal{A}}$-*entropy*.

**Definition 3.1.** ($H_{\ell,\mathcal{A}}$-entropy of $f$). Given a prior distribution $p(f)$ on functions, and a dataset $\mathcal{D}$ of observed function evaluations, the posterior $H_{\ell,\mathcal{A}}$-entropy with loss $\ell$ and action set $\mathcal{A}$ is defined as

$$H_{\ell,\mathcal{A}}\left[f \mid \mathcal{D}\right] = \inf_{a \in \mathcal{A}} \mathbb{E}_{p(f|\mathcal{D})}\left[\ell(f, a)\right]. \tag{1}$$

Intuitively, after expending our budget of function queries, suppose that we must make a terminal action $a^* \in \mathcal{A}$, where this action incurs a loss $\ell(f, a^*)$ defined by the loss $\ell$ and function $f$. Given a posterior $p(f|\mathcal{D})$ that describes our belief about $f$ after observing $\mathcal{D}$, we take the terminal action $a^*$ to be the *Bayes action*, i.e. the action that minimizes the posterior expected loss, $a^* = \arg\inf_{a \in \mathcal{A}} \mathbb{E}_{p(f|\mathcal{D})}\left[\ell(f, a)\right]$. The $H_{\ell,\mathcal{A}}$-entropy can then be viewed as the posterior expected loss of the Bayes action. We next describe how this generalizes Shannon entropy, and why it is a reasonable definition for an uncertainty measure.

**Example: Shannon entropy**  Let $\mathcal{P}(\mathcal{F})$ denote a set of probability distributions on a function space $\mathcal{F}$, which we assume contains the posterior distribution $p(f \mid \mathcal{D}) \in \mathcal{P}(\mathcal{F})$. Suppose, for the $H_{\ell,\mathcal{A}}$-entropy, that we let the action set $\mathcal{A} = \mathcal{P}(\mathcal{F})$, and loss function $\ell(f, a) = -\log a(f)$, for $a \in \mathcal{P}(\mathcal{F})$. Unlike the previous examples, note that the action set is now a set of distributions.

Then, the Bayes action will be $a^* = p(f \mid \mathcal{D})$ (this can be shown by writing out the definition of the Bayes action as a cross entropy, see Appendix A.1), and thus

$$H_{\ell,\mathcal{A}}[f \mid \mathcal{D}] = \mathbb{E}_{p(f|\mathcal{D})}\left[-\log a^*(f)\right] = H[f \mid \mathcal{D}], \tag{2}$$

where $H[f \mid \mathcal{D}] = -\int p(f \mid \mathcal{D}) \log p(f \mid \mathcal{D})$ is the Shannon differential entropy. Thus, the $H_{\ell,\mathcal{A}}$-entropy using the above $(\ell, \mathcal{A})$ is equal to the Shannon differential entropy.

Note that we have focused here on the Shannon entropy of the posterior over functions $p(f \mid \mathcal{D})$. In Section 4 we show how this example can be extended to the Shannon entropy of the posterior over properties of $f$, such as the location (or values) of optima, which will provide a direct equivalence to entropy search methods in BO.

**Why is this a reasonable measure of uncertainty?**  The $H_{\ell,\mathcal{A}}$-entropy has been interpreted as a measurement of uncertainty in the literature because it satisfies a few intuitive properties. First, similar to Shannon differential entropy, the $H_{\ell,\mathcal{A}}$-entropy is a *concave uncertainty measure* [13, 21]. Intuitively, if we have two distributions $p_1$ and $p_2$, and flip a coin to sample from $p_1$ or $p_2$, then we should have less uncertainty if we were told the outcome of the coin flip than if we weren't. In other words, the average uncertainty of $p_1$ and $p_2$ (i.e. coin flip outcome *known*) should be less than the uncertainty of $0.5p_1 + 0.5p_2$ (coin flip outcome *unknown*). Since $H_{\ell,\mathcal{A}}$ is concave, it has this property. As a consequence—also similar to Shannon differential entropy—the $H_{\ell,\mathcal{A}}$-entropy of the posterior is less than the $H_{\ell,\mathcal{A}}$-entropy of the prior, in expectation. Intuitively, whenever we make additional observations (i.e. gain more information), the posterior entropy is expected to decrease.

**Acquisition function**  We propose a family of acquisition functions for BO based on the $H_{\ell,\mathcal{A}}$-entropy, which are similar in structure to information-theoretic acquisition functions in the entropy search family. Like these, our acquisition function selects the query $x_t \in \mathcal{X}$ that maximally reduces the uncertainty, as characterized by the $H_{\ell,\mathcal{A}}$-entropy, in expectation. We refer to this quantity as the *expected $H_{\ell,\mathcal{A}}$-information gain* (EHIG).

**Definition 3.2.** (Expected $H_{\ell,\mathcal{A}}$-information gain). Given a prior $p(f)$ on functions and a dataset of observed function evaluations $\mathcal{D}_t$, the expected $H_{\ell,\mathcal{A}}$-information gain (EHIG), with loss $\ell$ and action set $\mathcal{A}$, is defined as

$$\text{EHIG}_t(x; \ell, \mathcal{A}) = H_{\ell,\mathcal{A}}\left[f \mid \mathcal{D}_t\right] - \mathbb{E}_{p(y_x|\mathcal{D}_t)}\left[H_{\ell,\mathcal{A}}\left[f \mid \mathcal{D}_t \cup \{(x, y_x)\}\right]\right]. \tag{3}$$

There are multiple benefits to developing this acquisition function. Though similar in form to entropy search acquisition functions, the EHIG yields (based on the definition of $H_{\ell,\mathcal{A}}$) the one-step Bayes

optimal query for the associated decision problem specified by the given loss $\ell$ and action set $\mathcal{A}$. We prove in Section 4 that the EHIG casts both uncertainty-reduction and decision-theoretic acquisition functions under a common umbrella, using different choices of $\ell$ and $\mathcal{A}$; this standardization provides guidance on which acquisition function is optimal for a given use case, based on details of the associated terminal action. More interestingly, in Section 5 we show how the EHIG allows us to derive problem-specific acquisition functions tailored to novel optimization and sequential decision making tasks. And importantly, since we frame acquisition optimization of this family in a common way—as a bilevel optimization problem over the sample space and action space—we can develop a single acquisition optimization method that can generically apply to many custom tasks (Section 6).

In Algorithm 1, we present $H_{\ell,\mathcal{A}}$-ENTROPY SEARCH, our full Bayesian optimization procedure using the EHIG acquisition function. This procedure takes as input a loss $\ell$, action set $\mathcal{A}$, and prior model $p(f)$. At each iteration, the procedure optimizes $\text{EHIG}_t(x; \ell, \mathcal{A})$ to select a design $x_t \in \mathcal{X}$ to query, and then evaluates the black-box function on this design to observe an outcome $y_{x_t} \sim \mathbf{f}(x_t) + \epsilon$. In Section 6 we describe methods for optimizing the EHIG acquisition function via gradient-based procedures, which provide a computationally efficient algorithm for many $\mathcal{A}$ and $\ell$.

---

**Algorithm 1** $H_{\ell,\mathcal{A}}$-ENTROPY SEARCH

---

**Input:** initial dataset $\mathcal{D}_1$, prior $p(f)$, action set $\mathcal{A}$, loss $\ell$.
 1: **for** $t = 1, \ldots, T$ **do**
 2:      $x_t \leftarrow \arg\max_{x \in \mathcal{X}} \text{EHIG}_t(x; \ell, \mathcal{A})$          ▷ Optimize the EHIG acquisition function
 3:      $y_{x_t} \sim \mathbf{f}(x_t) + \epsilon$          ▷ Evaluate the function $\mathbf{f}$ at $x_t$
 4:      $\mathcal{D}_{t+1} \leftarrow \mathcal{D}_t \cup \{(x_t, y_{x_t})\}$          ▷ Update the dataset
**Output:** distribution $p(f \mid \mathcal{D}_{T+1})$

---

## 4 A Unified View of Information-based and Decision-theoretic Acquisitions

In this section, we aim to show how acquisition functions commonly used in BO are special cases of the proposed EHIG family, for particular choices of $\ell$ and $\mathcal{A}$. This will allow us to view each acquisition function (including information-based ones) from the perspective of a common decision problem: after the BO procedure is complete, we choose a terminal action from action set $\mathcal{A}$ and then incur a loss defined by $\ell$. Each acquisition function can be viewed as reducing the posterior uncertainty over $f$ in a way that yields a terminal action with lowest expected loss.

This unified view provides two main benefits. First, it sheds light on the particular scenarios in which one of the existing acquisition functions is optimal over the others (which we focus on in this section). Second, it shows how using the EHIG with other choices for $\ell$ and $\mathcal{A}$ provides new acquisition functions for a broader set of optimization scenarios and related tasks (which is the focus of Section 5).

**Information-based acquisition functions** We state the family of entropy search acquisitions function in a general way that includes the entropy search (ES) [22], predictive entropy search (PES) [24], and max-value entropy search (MES) [47] algorithms. Let $\theta_f \in \Theta$ denote a property of $f$ we would like to infer. For example, we could set $\theta_f = \arg\max_{x \in \mathcal{X}} f(x) = x^* \in \mathcal{X}$, i.e. the location of the global maximizer of $f$, or $\theta_f = \max_{x \in \mathcal{X}} f(x) \in \mathbb{R}$, i.e. the maximum value achieved by $f$ in $\mathcal{X}$. This family of entropy search acquisition function can then be written as:

$$\text{ES}_t(x) = H\left[\theta_f | \mathcal{D}_t\right] - \mathbb{E}_{p(y_x | \mathcal{D}_t)}\left[H\left[\theta_f | \mathcal{D}_t \cup \{(x, y_x)\}\right]\right].$$

We can view this acquisition function as a special case of the EHIG in the following way. Suppose, after the BO procedure is complete, we choose a distribution $q$ from a set of distributions $\mathcal{P}(\Theta)$ and then incur a loss equal to the negative log-likelihood of $q$ for the true value of $\theta_f$. In this case, we view the action set as $\mathcal{A} = \mathcal{P}(\Theta)$ and the loss function as $\ell(f, a) = -\log a(\theta_f)$, where $a \in \mathcal{A}$. To visualize this, in the case where $\theta_f = x^*$, see Figure 1 (left), which shows the terminal action (gold density function) and corresponding loss (horizontal dashed line).

Under this choice, the $H_{\ell,\mathcal{A}}$-entropy of $f$ will be equal to the Shannon entropy of $\theta_f$, and thus the $\text{EHIG}_t$ will be equal to $\text{ES}_t$. We formalize this in the following proposition.

*Proposition* 1. If we choose $\mathcal{A} = \mathcal{P}(\Theta)$ and $\ell(f, q) = -\log q(\theta_f)$, then the EHIG is equivalent to the entropy search acquisition function, i.e. $\text{EHIG}_t(x; \ell, \mathcal{A}) = \text{ES}_t(x)$.

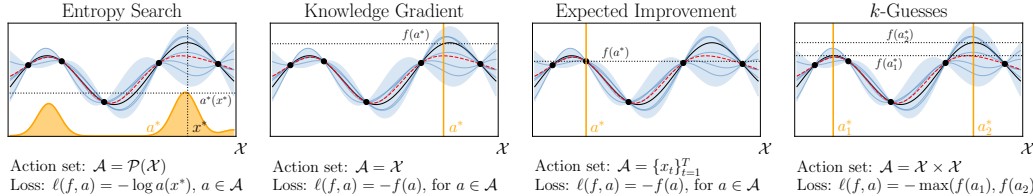

Figure 1: Example acquisition functions, and their corresponding Bayes actions $a^*$ visualized. For each, we write the associated action set $\mathcal{A}$ and loss function $\ell$ below the plot. In each plot, the true function is a solid black line, the posterior mean is a red dashed line, the observed data are black dots, and the Bayes action is shown in gold. See Section 4 for further discussion.

*Proof of Proposition 1.* See Appendix A.1. □

**Decision-theoretic acquisition functions**  We next describe how the EHIG generalizes decision-theoretic acquisition functions such as knowledge gradient (KG) and expected improvement (EI). Since these acquisition functions are often motivated from a perspective of a terminal decision, it is straightforward to show how they are a special case of the EHIG. However, the choice of $\mathcal{A}$ and $\ell$ here is insightful to review before extending EHIG to other scenarios.

First, the KG acquisition function can be written

$$\text{KG}_t(x) = \mathbb{E}_{p(y_x|\mathcal{D}_t)}\left[\mu_{t+1}^*(x, y_x)\right] - \mu_t^*,$$

where $\mu_t^* = \sup_{x' \in \mathcal{X}} \mathbb{E}_{p(f|\mathcal{D}_t)}\left[f(x')\right]$ is the max value of the posterior mean of $f$ given data $\mathcal{D}_t$, and $\mu_{t+1}^*(x, y_x) = \sup_{x' \in \mathcal{X}} \mathbb{E}_{p(f|\mathcal{D}_t \cup \{(x,y_x)\})}\left[f(x')\right]$ is the max value of the posterior mean, given both data $\mathcal{D}_t$ and observation $(x, y_x)$. Second, the EI acquisition function can be written

$$\text{EI}_t(x) = \mathbb{E}_{p(y_x|\mathcal{D}_t)}\left[\max(y_x - f_t^*, 0)\right],$$

where $f_t^* = \max\{\hat{f}(x_i)\}_{i=1}^{t-1}$, for $x_i \in \mathcal{D}_t$ and $\hat{f}(x_i)$ is the posterior expected value of $f$ at $x_i$. This definition is equal to the standard formulation of EI in the noiseless setting (i.e. when $y_x = \mathbf{f}(x)$ for queried $x$) and the *plug-in* formulation of EI in the noisy setting, when $y_x = \mathbf{f}(x) + \epsilon$ [37, 6].

To view these decision-theoretic acquisition functions as special cases of the EHIG, suppose that after BO is complete, we aim to make a single guess $x^* \in \mathcal{A}$ for the maximizer of $f$, and then incur a loss equal to the value of the function at $x^*$, i.e. $\ell(f, x^*) = -f(x^*)$. For KG, we let $\mathcal{A} = \mathcal{X}$, in which case the Bayes action $a^*$ is the maximizer of the posterior mean, and for EI, we let $\mathcal{A} = \{x_i\}_{i=1}^{t-1}$, in which case $a^*$ is the best queried point. We visualize these as gold vertical lines in Figure 1 (center panels). In Appendix A, we prove this equivalence with $\text{EHIG}_t$, for $\text{KG}_t$ and $\text{EI}_t$. Additionally, in Appendix A.4 we discuss connections to the probability of improvement (PI) acquisition function.

We thus see two key differences here, in comparison with information-based BO: (i) the terminal action $a^*$ is a point estimate of the optimizer $x^*$ rather than a distribution over $\mathcal{X}$, and (ii) the loss does not depend on the particular value of the true optima $x^*$ (nor on how accurately $a^*$ provides an estimate of $x^*$), but rather only depends on the function value of the terminal action, $f(a^*)$.

# 5 A Framework to Derive New Acquisition Functions for Custom Tasks

There are many real-world problems that go beyond simple black-box optimization, which have not been explicitly studied in the literature, and for which there does not exist a suitable acquisition function. For these use cases, we can define an action set and loss based on details of the problem, and use the EHIG to provide a corresponding problem-specific acquisiton function. As examples of this, below we apply the EHIG to a number of relevant problems where, as far as we are aware, no corresponding acquisition function has been developed by prior work.

**Illustrative example: $k$-guesses**  As a simple example to illustrate our framework, suppose, after optimization is complete, we are allowed to make a batch of $k$ guesses for the function maximizer $x^*$, and then recieve a final reward based on the best guess. This setup appears in cases where, after BO is complete, we can make a *batch* of final designs (e.g. synthesize a final set of materials [44] or train a final set of models [49, 41]), and only care about the single *best* design of the batch. We can thus view the action set as $\mathcal{A} = \mathcal{X}^k$, and loss as $\ell(f, a) = -\max\left(f(a_1), \ldots, f(a_k)\right)$.

Figure 1 (far right) provides a visualization of this for $k = 2$. In this scenario, one of the Bayes actions ($a_2^*$) is near the maximizer of the posterior mean (similar to KG), while the other ($a_1^*$) is separated from the first. Intuitively, to minimize the expected loss, the second guess should have both a high posterior mean, and also a low correlation to the first guess—the first guess has a better chance of a low loss, but in cases where it fails, we want the second guess to *not* fail (i.e. not match the first guess), while also achieving a low loss. In practice, this yields an EHIG acquisition function that has a similar but distinct exploration strategy from KG. It spends a small portion of the budget on queries that give some information about not just the first guess, but the other $k$-1 guesses as well.

**Top-$k$ optimization with diversity**   Instead of a single optimal point in $\mathcal{X}$, there are applications where we wish to estimate a set of top-$k$ optima, i.e. the subset of $\mathcal{X}$ of size $k$ that has the highest sum of values under $f$. Examples of this can be found in materials discovery [30, 44], sensor networks [2, 18], and medicine [51]. Note that the goal of this problem is distinct from the $k$-guesses example described above. When $\mathcal{X}$ is continuous, to avoid redundant solutions, we may wish to carry out the task of *top-k optimization with diversity*, which aims to find the top-$k$ optima such that $d(x_i, x_j) \geq c$, $\forall i, j \in \{1, \ldots, k\}$, for a problem-specific distance $d$. As one example of our EHIG framework, suppose that we choose $\mathcal{A} = \mathcal{X}^k$ (where $a = (a_1, \ldots, a_k) \in \mathcal{A}$ denotes a set of top-$k$ points) and incur the loss

$$\ell(f, a) = -\sum_i f(a_i) - \sum_{1 \leq i < j \leq k} d(a_i, a_j). \tag{4}$$

Note that we can select the distance function $d$ here to match the problem-specific constraint. Intuitively speaking, this choice of $(\ell, \mathcal{A})$ yields an EHIG acquisition function that makes a sequence of queries which rotate focus between multiple diverse optimal points in the domain.

**Multi-level set estimation**   The goal of level set estimation (LSE) is to estimate a subset of the design space $\mathcal{X}$, where function values are larger than a given threshold $c$, $\mathcal{S}_c = \{x \in \mathcal{X} : f(x) > c\}$. This task appears in a number of applications, including catalyst design [54], interactive learning [5], and environmental monitoring [40]. While many prior works have studied standard LSE, here we consider the task of multi-level set estimation (MLSE), where we are given $m$ thresholds satisfying $c_1 < \ldots < c_m$ and want to estimate $m + 1$ sets: $\mathcal{S}_i = \{x \in \mathcal{X} : c_i < f(x) < c_{i+1}\}$ for $i = \{0, \ldots, m\}$ (where $c_0 := -\infty$ and $c_{m+1} := +\infty$). This is useful in the above applications when we have more than one threshold of interest—for example, public health policy makers must estimate regions where disease prevalence exceeds 1%, 2%, etc., for graded reopening decisions [36, 52].

As one approach to MLSE using the EHIG, we focus on settings with a discrete set of design points $\mathcal{X}_0 \subset \mathcal{X}$, $|\mathcal{X}_0| = J$ [19, 26]. Suppose, after querying is complete, we must choose a set of values $a \in \mathcal{A} = [0, 1]^{J \times m}$ (one for each $x \in \mathcal{X}_0$ and $i \in \{0, \ldots, m\}$), which represent level set identity variables. Suppose we then incur a loss with the following form, that depends on these identity variables $a$, as well as on a flexible relation $r(f(x), c_i)$ between function values $f(x)$ and thresholds $c_i$, i.e.

$$\ell(f, a(x)) = -\sum_{i=1}^{m} \sum_{x \in \mathcal{X}_0} a_i(x) r(f(x), c_i). \tag{5}$$

For instance, if $r(f(x), c_i) = f(x) - c_i$, then the optimal $a_i(x)$ should specify the $c_i$-super level set for each $i \in \{1, \ldots, m\}$, i.e. $a_i(x) = 1$ for each $x \in \mathcal{X}_0$ with $f(x) > c_i$ and $a_i(x) = 0$ otherwise. This example loss yields an acquisition function that, empirically, focuses samples around the boundaries of multiple level sets of a black-box function simultaneously.

**Sequence search**   We define *sequence search* as the task of estimating a sequence of inputs $(x_1, \ldots, x_m) \in \mathcal{X}^m$ with outputs values matching a set of problem-specific criteria. For example, we may wish to estimate a sequence of inputs corresponding to a predefined set of function values $(y_1^\circledast, \ldots, y_m^\circledast)$. This finds applications in materials science, such as in the task of synthesizing a nanoparticle library [15] (i.e. finding a set of input conditions that yield a set of nanoparticles of different pre-defined sizes). As another example, in the context of public health, we may be interested in a set of locations where vaccination rates equal some pre-specified values (e.g. $(20\%, \ldots, 80\%)$) when making decisions involving vaccine allocations, as we describe in Section 7. As an example of these applications in our EHIG framework, we might have the action set $\mathcal{A} = \mathcal{X}^m$ and loss

$$\ell(f, \mathbf{a}) = \sum_{i=1}^{m} (f(\mathbf{a}_m) - y_m^{\circledast})^2. \tag{6}$$

These examples all aim to show that the EHIG can be used to define a problem-specific acquisition function, which can be tailored to the details of a particular use case. As a result, when used in Algorithm 1, we gain a customizable optimization framework that can be applied to a variety of novel problem settings with special-purpose losses.

## 6  Gradient-based Acquisition Optimization

At each iteration of $H_{\ell,\mathcal{A}}$-ENTROPY SEARCH (Algorithm 1), we optimize the acquisition function to select the next query $x_t = \arg\max_{x \in \mathcal{X}} \mathrm{EHIG}_t(x; \ell, \mathcal{A})$. Classically, zeroth order optimization routines have been used for acquisition optimization in BO. However, recent work has developed gradient-based methods for optimizing certain acquisition functions [48, 4], which can allow for efficient acquisition optimization over $\mathcal{X}$. We work on similar methodology here—namely, we develop a gradient-based acquisition optimization procedure for appropriate settings (i.e. assuming continuous $\mathcal{X}$ and $\mathcal{A}$, and certain conditions on $\ell$). We can, for example, apply this gradient-based optimization to each of the acquisition functions described in Section 5, for which we show experimental results in Section 7.

Similar to related work [48, 4], we give the following derivation with a focus on Gaussian process (GP) models, though the methodology can be extended to other models in which we can apply the reparameterization procedure described below to differentiate through posterior model parameters.

**Differentiable loss function**  We first describe a few assumptions that must be satisfied to carry out the gradient-based optimization procedure. Denote the posterior expected loss given $\mathcal{D}$ by $L(\mathcal{D}, a) := \mathbb{E}_{p(f|\mathcal{D})}[\ell(f, a)]$. We assume that this loss function depends only on the function value of $f$ at a finite number of points, i.e. there exists functions $\mathfrak{x}_1(a), \cdots, \mathfrak{x}_K(a)$, and a function $\ell' : \mathbb{R}^K \times \mathcal{A} \to \mathbb{R}$, for $K \in \mathbb{N}$, such that

$$\ell(f, a) = \ell'(f(\mathfrak{x}_1(a)), f(\mathfrak{x}_2(a)), \cdots, f(\mathfrak{x}_K(a)), a). \tag{7}$$

This requirement is satisfied by the loss functions in Section 5. For brevity, denote the sequence $\mathfrak{x}_1(a), \cdots, \mathfrak{x}_K(a)$ by $\mathfrak{x}_{1:K}(a)$ and $f(\mathfrak{x}_1(a)), \cdots, f(\mathfrak{x}_K(a))$ by $f(\mathfrak{x}_{1:K}(a))$. We assume that the functions $\mathfrak{x}_k$ and $\ell'$ are differentiable with respect to all arguments. Given a dataset $\mathcal{D}$ and GP prior, the posterior distribution of $f(\mathfrak{x}_K(a))$ is also Gaussian. In particular, there exist functions

$$\mu : \mathfrak{x}_{1:K}(a) \times \mathcal{D} \mapsto \mathbb{R}^K \quad \text{and} \quad U : \mathfrak{x}_{1:K(a)} \times \mathcal{D} \mapsto \mathbb{R}^{K \times K}$$

such that $f(\mathfrak{x}_{1:K}(a)) = \mu(\mathfrak{x}_{1:K}(a); \mathcal{D}) + U(\mathfrak{x}_{1:K}(a); \mathcal{D})\epsilon$ where $\epsilon$ is drawn from a $K$-dimensional standard normal distribution. We can combine the above results to get

$$L(\mathcal{D}, a) = \mathbb{E}_\epsilon \left[ \ell'(\mu(\mathfrak{x}_{1:K}(a); \mathcal{D}) + U(\mathfrak{x}_{1:K}(a); \mathcal{D})\epsilon, a) \right].$$

A key property is that we can compute unbiased gradients of this with respect to both $\mathcal{D}$ and $a$, as

$$\nabla L(\mathcal{D}, a) = \mathbb{E}_\epsilon \left[ \nabla \ell'(\mu(\mathfrak{x}_{1:K}(a); \mathcal{D}) + U(\mathfrak{x}_{1:K}(a); \mathcal{D})\epsilon, a) \right].$$

**Differentiable acquisition function**  For a given input $x \in \mathcal{X}$, let $y(x, \mathcal{D})$ denote the posterior predictive distribution of our model. Note that there exists a deterministic function $\bar{y}(x, \mathcal{D}, \lambda)$ such that $y(x, \mathcal{D}) = \bar{y}(x, \mathcal{D}, \lambda)$, where $\lambda$ is drawn from a standard normal distribution. Hence, if $\ell$ satisfies Eq. (7), then we can optimize $\mathrm{EHIG}_t$ with gradient descent. In particular, we can write

$$\inf_{x \in \mathcal{X}} -\mathrm{EHIG}_t(x; \ell, \mathcal{A}) = \inf_{x \in \mathcal{X}} \inf_{\mathfrak{a}: \lambda \mapsto \mathcal{A}} \mathbb{E}_{\lambda, \epsilon}[\ell'(\hat{\mu}(x, \mathfrak{a}(\lambda)) + \hat{U}(x, \mathfrak{a}(\lambda))\epsilon, \mathfrak{a}(\lambda))] \tag{8}$$

where in Eq. (8), to avoid clutter, we use the shorthand $\hat{\mu}(x, \mathfrak{a}(\lambda)) := \mu(\mathfrak{x}_{1:K}(\mathfrak{a}(\lambda)); \mathcal{D} \cup \bar{y}(x, \mathcal{D}, \lambda))$, and $\hat{U}(x, \mathfrak{a}(\lambda)) := U(\mathfrak{x}_{1:K}(\mathfrak{a}(\lambda)); \mathcal{D} \cup \bar{y}(x, \mathcal{D}, \lambda))$. Importantly, we can compute the unbiased gradient of the quantity $\mathbb{E}_{\lambda, \epsilon}[\ell'(\hat{\mu}(x, \mathfrak{a}(\lambda)) + \hat{U}(x, \mathfrak{a}(\lambda))\epsilon, \mathfrak{a}(\lambda))]$. In practice, we can also take gradients of a Monte Carlo estimate of Eq. (8) [4], by fixing samples of $\lambda, \epsilon$ throughout the optimization. Specifically, we can sample $\lambda_1, \cdots, \lambda_M$ and $\epsilon_1, \cdots, \epsilon_N$ and approximate Eq. (8) via

$$\inf_{x \in \mathcal{X}} -\mathrm{EHIG}_t(x; \ell, \mathcal{A}) \approx \inf_{x \in \mathcal{X}} \inf_{a_1, \ldots, a_M} \frac{1}{NM} \sum_{m,n} \ell'(\hat{\mu}(x, a_m) + \hat{U}(x, a_m)\epsilon_n, a_m), \tag{9}$$

where we use $a_m = a(\lambda_m)$ for brevity. Under the assumptions above, we can compute the unbiased gradient of this quantity, and by using systems such as GPyTorch [17] and BoTorch [4] we can compute this gradient efficiently via automatic differentiation.

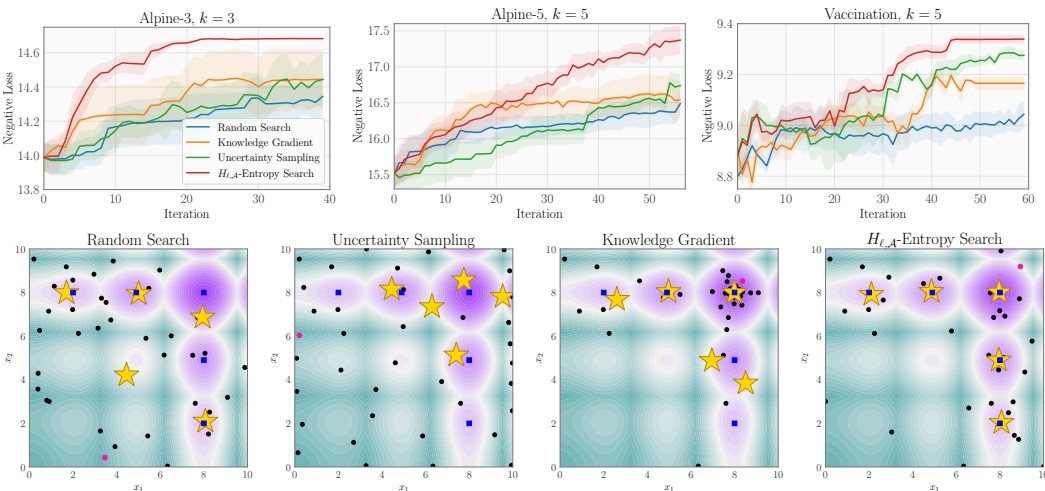

Figure 2: **Top-$k$ optimization with diversity.** *Top row:* Plots of the negative loss $-\ell(\mathbf{f}, a^*)$ versus iteration for all methods, on the *Alpine-3*, *Alpine-5* and *Vaccination* functions, where error bars represent one standard error. *Bottom row:* Visualization of methods in two dimensions, showing the set of ground-truth top-5 diverse design points (blue squares), queries $\mathcal{D}_t$ taken (black dots), acquisition function optimizer (pink dot), and the estimated set of top-5 diverse design points (gold stars).

## 7  Experiments

We evaluate our proposed method on the example tasks described in Section 5: top-$k$ optimization with diversity, multi-level set estimation, and sequence search. For these applications, we show comparisons against a set of baselines on real and synthetic black-box functions.

**Comparison methods.** In our experiments, we compare the following set of acquisition strategies:

- $H_{\ell,\mathcal{A}}$-ENTROPY SEARCH (HES). We follow Algorithm 1, using the loss and action set for each task as described in Section 5, and the gradient-based procedure outlined in Section 6.
- RANDOM SEARCH (RS). At each iteration, we draw a sample $x_t$ uniformly at random from $\mathcal{X}$.
- UNCERTAINTY SAMPLING (US). At each iteration, we select the point that maximizes the posterior predictive variance, i.e. $x_t = \arg\max_{x \in \mathcal{X}} \mathrm{Var}[p(y_x \mid \mathcal{D}_t)]$.
- KNOWLEDGE GRADIENT (KG). We show KG as a representative method for standard BO. KG allows us to carry out a similar gradient-based procedure as in HES.
- PROBABILITY OF MISCLASSIFICATION (POM). This is a common acquisition function for level set estimation [7]. We predict whether a point is above a threshold, represented by a binary variable $z$, and select the design with maximal label uncertainty $x_t = \arg\min_{x \in \mathcal{X}} \max_{z \in \{0,1\}} p(z|x)$.

Note that we are restricted to comparing against relatively general-purpose baseline methods, as more-specific acquisition functions have not previously been developed for the tasks below.

**Top-$k$ Optimization with Diversity**  In our first task, the goal is to find a set of $k$ diverse elements in $\mathcal{X}$, each with a high value of $\mathbf{f}$. To assess each method, at each iteration we record the negative loss $-\ell(\mathbf{f}, a^*)$ using Eq. (4)—i.e. the *negative top-$k$ with diversity loss* of the Bayes action $a^* = \arg\inf_{a \in \mathcal{A}} \mathbb{E}_{p(f|\mathcal{D}_t)}[\ell(f, a)]$ on the true function $\mathbf{f}$—using the set of queries $\mathcal{D}_t$ produced by the given method. Intuitively, if a method makes a set of queries that yield a good estimate of diverse top-$k$ elements, it will score a high value under this metric.

In Figure 2 (*bottom row*) we visualize results on the multimodal *Alpine-d* benchmark function (see appendix for details). Here, we can see that HES concentrates queries over five local optima of this function, while KG allocates a majority of samples on only the highest peak, and both US and RS distribute their queries over the full domain $\mathcal{X}$. In Figure 2 (*top row*), we compare performance by plotting the negative loss versus iteration on two higher dimensional examples. We also compare each method on the *Vaccination* function (provided by [53]), which returns the vaccination rate for locations in the continental United States, given an input *(latitude, longitude)*. The goal of this task is to efficiently find a set of five diverse locations with highest vaccination rates. We show results in Figure 2 (*top row, right*), and see a similar advantage of HES over comparison methods.

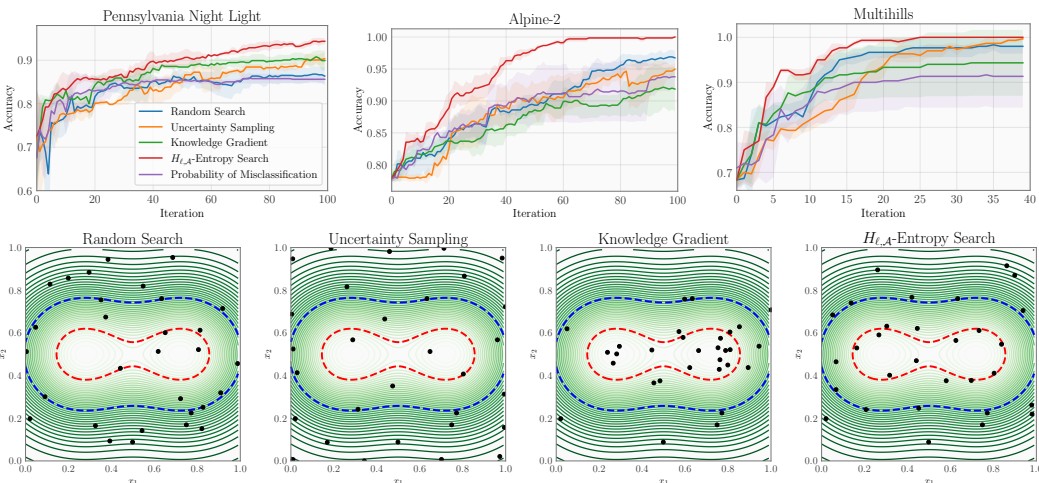

Figure 3: **Multi-level set estimation.** *Top row*: Plots of accuracy versus iteration for all methods, where error bars represent one standard error. *Bottom row:* Visualization of methods on the *Multihills* function, showing the ground-truth level set boundaries (red and blue dashed lines) and queries $\mathcal{D}_t$ taken (black dots).

**Multi-level Set Estimation**    In our second task, the goal is to carry out multi-level set estimation. Here, we can assess each method using a more conventional metric: we produce an estimate of the level for every $x \in \mathcal{X}_0$, using the model's posterior mean (given the queries selected by a particular method), and then record the accuracy of this estimate averaged across all level set thresholds. Intuitively, a method will achieve a higher accuracy if it chooses queries that yield a better estimate of the function near the threshold boundaries of the level sets. In Figure 3 (*bottom row*), we visualize results for a two-level set task on the *Multihills* function, defined as a mixture density (details given in appendix). We see that HES concentrates queries along both of the boundaries, which are drawn as blue and red dashed lines. In the *top row*, we compare the performance of all methods, showing the accuracy vs. iteration. Here, the *Pennsylvania Night Light* function [1] released by NASA (additional details in the appendix), returns the relative level of light at a location in Pennsylvania, as queried by a satellite image. The goal of this experiment is to determine the portion of land at which night light is above a specified threshold value. In Appendix B, we show additional experiments, including a visualization of results on this function.

**Sequence Search**    In our third task, the goal is to find a sequence of elements whose value under the black-box function matches a set of pre-specified function values $(y_1^{\circledast}, \ldots, y_m^{\circledast})$. To assess each method, at each iteration we record the negative loss $-\ell(\mathbf{f}, a^*)$ from Eq. (6)—i.e. the negative *sequence search loss* of the Bayes action $a^* = \arg\inf_{a \in \mathcal{A}} \mathbb{E}_{p(f|\mathcal{D}_t)}[\ell(f, a)]$— using the set of queries $\mathcal{D}_t$ produced by the given method. Intuitively, if a method makes a set of queries that yield a good estimate of a sequence of $(x_1^{\circledast}, \ldots, x_m^{\circledast})$ such that $(\mathbf{f}(x_1^{\circledast}), \ldots, \mathbf{f}(x_m^{\circledast})) \approx (y_1^{\circledast}, \ldots, y_m^{\circledast})$, it will score a high value on this metric.

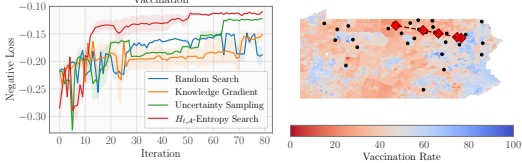

Figure 4: **Sequence search.** *Left:* Negative loss versus iteration, where error bars represent one standard error. *Right:* Visualization of the *Vaccination* function, along with the queries $\mathcal{D}_t$ taken by HES (black dots), and the estimated sequence $(x_1^{\circledast}, \ldots, x_5^{\circledast})$ (red diamonds), such that $(f(x_1^{\circledast}), \ldots, f(x_5^{\circledast})) = (30\%, 40\%, 50\%, 60\%, 70\%)$.

In Figure 4 (*right*) we visualize results on the *Vaccination* function (described above). Here, our goal is to find a sequence of five *(latitude, longitude)* coordinates with vaccination rates equal to $(y_1^{\circledast}, \ldots, y_m^{\circledast}) = (30\%, 40\%, 50\%, 60\%, 70\%)$. Estimates of locations that match these function values can be useful when making policy decisions involving a vaccine response or allocation. In this case, we see that HES concentrates queries along a route from the relatively highly vaccinated region in the East to the relatively lowly vaccinated region in the North. The *left* plots in Figure 4 provides a quantitive comparison of methods on the *Vaccination* function (results on additional functions are shown in the appendix), plotting the negative loss vs. iteration. These again show the benefits of query selection performed by HES relative to the comparison strategies.

# 8 Conclusion

In this paper, we take a decision making perspective on information-based acquisition functions: after querying is complete, we assume that we must make some decision $a^*$ and then incur a loss $\ell(\mathbf{f}, a^*)$. Our goal is thus to make a sequence of queries that reduce the uncertainty of the posterior distribution $p(f \mid \mathcal{D}_t)$ in a way to best help make this decision with low loss. Using $H_{\ell, \mathcal{A}}$-entropy [13, 39], we can define an EHIG acquisition function which carries this out directly: it selects a point that is expected to maximally reduce the posterior expected loss of the Bayes action $a^*$. We incorporate this acquisition function into a procedure called $H_{\ell, \mathcal{A}}$-ENTROPY SEARCH, and show, in many cases, that we can perform efficient gradient-based optimization of this acquisition function.

There are multiple interesting avenues for future work. First, we hope to develop acquisition optimization methods for additional settings, such as for non-continuous action sets $\mathcal{A}$ or design spaces $\mathcal{X}$ [12], and for functions with multidimensional outputs [23, 11]. One interesting avenue is hybrid optimization settings, where we can only take gradient steps with respect to either the design or action variables. Another potential direction is to incorporate cost-aware Bayesian optimization techniques into the EHIG framework [29, 50, 3]. We also wish to study how the proposed EHIG framework could be applied in practice to solve various problems in the sciences, including experimental physics [14, 32, 8], drug discovery [43, 27, 20], and materials design [31, 46]. Finally, we wish to study in further detail how the EHIG acquisition function could be implemented for Bayesian decision making with other probabilistic models beyond Gaussian processes [42, 10, 9, 34].

**Acknowledgments**

We thank the anonymous reviewers, members of the Stanford SAIL community, and members of the CMU Auton Lab for helpful feedback on this paper. This work was supported by NSF (#1651565), AFOSR (FA95501910024), ARO (W911NF-21-1-0125), CZ Biohub, and Sloan Fellowship.

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
