# A Proofs

Here we prove the propositions stated in Section 4.

## A.1 Entropy Search

**Proposition 1.** If we choose $\mathcal{A} = \mathcal{P}(\Theta)$ and $\ell(f, q) = -\log q(\theta_f)$, then the EHIG is equivalent to the entropy search acquisition function, i.e. $\mathrm{EHIG}_t(x; \ell, \mathcal{A}) = \mathrm{ES}_t(x)$.

*Proof of Proposition 1.* We first prove that under our definition of loss $\ell$, the $H_{\ell,\mathcal{A}}$-entropy $H[f \mid \mathcal{D}_t]$ is equivalent to the Shannon entropy of the posterior distribution over $\theta_f$ (where $\theta_f$ denotes a property of $f$ that we would like to infer—as an example, $\theta_f$ could be equal to the global maximizer $x^*$ of $f$).

Note that the $H_{\ell,\mathcal{A}}$-entropy is the expected loss of the Bayes action

$$q^* = \arg\inf_{q \in \mathcal{P}(\mathcal{X})} \mathbb{E}_{p(f|\mathcal{D}_t)} \left[ -\log q(\theta_f) \right].$$

We want to show that $q^*$ defined above is equal to $p(\theta_f \mid \mathcal{D}_t)$. To do so, note that

$$q^* = \arg\inf_{q \in \mathcal{P}(\mathcal{X})} \mathbb{E}_{p(f|\mathcal{D}_t)} \left[ -\log q(\theta_f|\mathcal{D}_t) \right] \tag{10}$$

$$= \arg\inf_{q \in \mathcal{P}(\mathcal{X})} \mathbb{E}_{p(\theta_f|\mathcal{D}_t)} \left[ -\log q(\theta_f|\mathcal{D}_t) \right] \tag{11}$$

$$= p(\theta_f|\mathcal{D}_t), \tag{12}$$

where the first equality holds since

$$E_X[f(g(X))] = E_Z[f(Z)], \text{ when } Z = g(X), \tag{13}$$

and the second equality holds since we can view $\mathbb{E}_{p(\theta_f|\mathcal{D}_t)} \left[ -\log q(\theta_f|\mathcal{D}_t) \right]$ as a cross entropy, which is minimized when $q(\theta_f|\mathcal{D}_t) = p(\theta_f|\mathcal{D}_t)$. Therefore, under this loss and action set, using the definition of the EHIG we can write

$$\mathrm{EHIG}_t(x; \ell, \mathcal{A}) = H\left[ p(\theta_f \mid \mathcal{D}_t) \right] - \mathbb{E}_{p(y_x|\mathcal{D}_t)} \left[ H\left[ p(\theta_f \mid \mathcal{D}_t \cup \{x, y_x\}) \right] \right] = \mathrm{ES}_t(x). \tag{14}$$

$\square$

## A.2 Knowledge Gradient

**Proposition 2.** If we choose $\mathcal{A} = \mathcal{X}$ and $\ell(f, x) = -f(x)$, then the EHIG is equivalent to the knowledge gradient acquisition function, i.e. $\mathrm{EHIG}_t(x; \ell, \mathcal{A}) = \mathrm{KG}_t(x)$.

*Proof of Proposition 2.* The proof follows directly from the definition of $H_{\ell,\mathcal{A}}$-entropy and the EHIG, namely

$$\mathrm{EHIG}_t(x) = \inf_{a \in \mathcal{A}} \mathbb{E}_{p(f|\mathcal{D}_t)} \left[ \ell(f, a) \right] - \mathbb{E}_{p(y_x|\mathcal{D}_t)} \left[ \inf_{a \in \mathcal{A}} \mathbb{E}_{p(f|\mathcal{D}_t \cup \{(x,y_x)\})} \left[ \ell(f, a) \right] \right] \tag{15}$$

$$= \inf_{x' \in \mathcal{X}} \mathbb{E}_{p(f|\mathcal{D}_t)} \left[ -f(x') \right] - \mathbb{E}_{p(y_x|\mathcal{D}_t)} \left[ \inf_{x' \in \mathcal{X}} \mathbb{E}_{p(f|\mathcal{D}_t \cup \{(x,y_x)\})} \left[ -f(x') \right] \right] \tag{16}$$

$$= -\sup_{x' \in \mathcal{X}} \mathbb{E}_{p(f|\mathcal{D}_t)} \left[ f(x') \right] + \mathbb{E}_{p(y_x|\mathcal{D}_t)} \left[ \sup_{x' \in \mathcal{X}} \mathbb{E}_{p(f|\mathcal{D}_t \cup \{(x,y_x)\})} \left[ f(x') \right] \right] \tag{17}$$

$$= \mathbb{E}_{p(y_x|\mathcal{D}_t)} \left[ \mu_{t+1}^*(x, y_x) \right] - \mu_t^* \tag{18}$$

$$= \mathrm{KG}_t(x) \tag{19}$$

$\square$

## A.3 Expected Improvement

**Proposition 3.** If we choose $\mathcal{A}_t = \{x_i\}_{i=1}^{t-1}$, where $x_i \in \mathcal{D}_t$, and $\ell(f, x_i) = -f(x_i)$, then the EHIG is equal to the expected improvement acquisition function, i.e. $\mathrm{EHIG}_t(x; \ell, \mathcal{A}) = \mathrm{EI}_t(x)$.

*Proof of Proposition 3.* The first term of $\text{EHIG}_t$ in Eq. (3) is equal to:

$$H_{\ell,\mathcal{A}_t}[f \mid \mathcal{D}_t] = \inf_{a \in \mathcal{A}_t} \mathbb{E}_{p(f|\mathcal{D}_t)}[\ell(f,a)] = -\max_{i \le t-1} \hat{f}(x_i) := -f_t^* \tag{20}$$

where $\hat{f}(x_i)$ is the posterior expected value of $f$ at $x_i$.

The second term in Eq. (3) is:

$$\mathbb{E}_{p(y_x|\mathcal{D}_t)}\left[H_{\ell,\mathcal{A}_{t+1}}[f \mid \mathcal{D}_t \cup \{(x,y_x)\}]\right] \tag{21}$$

$$= \mathbb{E}_{p(y_x|\mathcal{D}_t)}\left[\mathbb{E}_{p(f|\mathcal{D}_t \cup \{(x,y_x)\})}\left[\inf_{a \in A_{t+1}} \ell(f,a)\right]\right] \tag{22}$$

$$= \mathbb{E}_{p(y_x|\mathcal{D}_t)}\left[\mathbb{E}_{p(f|\mathcal{D}_t \cup \{(x,y_x)\})}[-\max(f_t^*, f(x))]\right] \tag{23}$$

$$= \mathbb{E}_{p(y_x|\mathcal{D}_t)}[-\max(f_t^*, y_x)] \tag{24}$$

Putting it together, the $\text{EHIG}_t$ acquisition function in Eq. (3) will reduce to:

$$\text{EHIG}_t(x; \ell, \mathcal{A}) = -f_t^* - \mathbb{E}_{p(y_x|\mathcal{D}_t)}[-\max(f_t^*, y_x)] \tag{25}$$

$$= \mathbb{E}_{p(y_x|\mathcal{D}_t)}[\max(0, y_x - f_t^*)] \tag{26}$$

$$= \text{EI}_t(x). \tag{27}$$

$\square$

## A.4 Probability of Improvement

We additionally include a result below showing that the probability of improvement (PI) acquisition function can similarly be viewed as a special case of the proposed EHIG family.

**Proposition 4.** For some constant $\tau$, the acquisition function of PI is defined as $\text{PI}_\tau(x; \mathcal{D}_t) = \mathbb{E}_{p(f|\mathcal{D}_t)}[\mathbb{I}(f(x) - \tau > 0)]$, where $\mathbb{I}(\cdot)$ is the indicator function, and typically $\tau$ is taken to be equal to $f_t^* = \max_{i \le t-1} \hat{f}(x_i)$ for $x_i \in \mathcal{D}_t$. If we choose $\mathcal{A}_t = \{x_{t-1}\}$, where $x_{t-1} \in \mathcal{D}_t$, and $\ell_\tau(f, x) = -\mathbb{I}(f(x) - \tau > 0)$, then maximizing EHIG is equivalent to maximizing the probability of improvement acquisition function, i.e. $\arg\max_{x \in \mathcal{X}} \text{EHIG}_t(x; \ell_\tau, \mathcal{A}) = \arg\max_{x \in \mathcal{X}} \text{PI}_\tau(x)$.

*Proof of Proposition 4.* The first term of $\text{EHIG}_t$ in Eq. (3) is equal to:

$$H_{\ell,\mathcal{A}_t}[f \mid \mathcal{D}_t] = \inf_{a \in \mathcal{A}_t} \mathbb{E}_{p(f|\mathcal{D}_t)}[\ell(f,a)] = -\mathbb{I}(\hat{f}(x_{t-1}) - \tau > 0) \tag{28}$$

where $\hat{f}(x_{t-1})$ is the posterior expected value of $f$ at $x_{t-1}$. More importantly, $H_{\ell,\mathcal{A}_t}[f \mid \mathcal{D}_t]$ is a constant with respect to $x$ that we are optimizing.

The second term in Eq. (3) is:

$$\mathbb{E}_{p(y_x|\mathcal{D}_t)}\left[H_{\ell,\mathcal{A}_{t+1}}[f \mid \mathcal{D}_t \cup \{(x,y_x)\}]\right] \tag{29}$$

$$= \mathbb{E}_{p(y_x|\mathcal{D}_t)}\left[\inf_{a \in \{x\}} \mathbb{E}_{p(f|\mathcal{D}_t \cup \{(x,y_x)\})}[\ell(f,a)]\right] \tag{30}$$

$$= \mathbb{E}_{p(y_x|\mathcal{D}_t)}\left[\mathbb{E}_{p(f|\mathcal{D}_t \cup \{(x,y_x)\})}[-\mathbb{I}(f(x) - \tau > 0)]\right] \tag{31}$$

$$= -\mathbb{E}_{p(y_x|\mathcal{D}_t)}[\mathbb{I}(y_x - \tau > 0)] \tag{32}$$

Putting it together, the $\text{EHIG}_t$ acquisition function in Eq. (3) will reduce to:

$$\text{EHIG}_t(x; \ell_\tau, \mathcal{A}) = -\mathbb{I}(\hat{f}(x_{t-1}) - \tau > 0) + \mathbb{E}_{p(y_x|\mathcal{D}_t)}[\mathbb{I}(y_x - \tau > 0)] \tag{33}$$

$$= \mathbb{E}_{p(y_x|\mathcal{D}_t)}[\mathbb{I}(y_x - \tau > 0)] + \text{constant} \tag{34}$$

$$= \text{PI}_\tau(x) + \text{constant}. \tag{35}$$

Thus maximizing EHIG is equivalent to maximizing the probability of improvement acquisition function.

$\square$

# B  Additional Experimental Details and Results

**Details on the *Alpine-d* function.**  The multimodal *Alpine-d* function is defined as $Alpine\text{-}d(x) = \sum_{i=1}^{d} |x_i \sin(x_i) + 0.1x_i|$, for $x \in \mathbb{R}^d$.

**Details on the *Vaccination* function.**  The vaccination function is obtained by training a Multi-Layer Perceptron (MLP) network based on the data from [53], which uses county-level vaccination data provided by the CDC, and uses small area estimation[3] to interpolate the vaccination rate of every location. We restrict the optimization domain to be a rectangle focusing on the state of Pennsylvania.

**Details on the *Multihills* function.**  The *Multihills* function is defined as a mixture density as follows. $Multihills(x) = \sum_{j=1}^{J} w_j \mathcal{N}(x \mid \mu_j, C_j)$, for $x \in \mathbb{R}^d$, where $\mathcal{N}$ denotes a multivariate normal density, $\{\mu_j\}$ are a set of $J$ means, $\{C_j\}$ are a set of J covariance matrices, and $\{w_j\}$ are a set of J weights.

**Details on the *Pennsylvania Night Light* function.**  We consider the 2012 gray scale global night-light raster with resolution 0.1 degree per pixel. The data is downloaded from NASA Earth Observatory[4]. We restrict the optimization domain to be a rectangle focusing on the state of Pennsylvania and normalize all raster data before use. Each location query gives a value proportional to the average amount of night light at that location.

**Computational Cost.**  While using the $\text{EHIG}_t(x; \ell, \mathcal{A})$ acquisition function in Bayesian optimization (Algorithm 1) is more expensive than simpler methods (e.g. expected improvement (EI)), in many cases it has a comparable computational cost to methods such as knowledge gradient (KG) or entropy search (ES) methods, when applied to the same task—in fact, our implementation has a similar structure as one-shot knowledge gradient acquisition optimization methods.

The following timing results compare the average cost (*mean wall clock time in seconds*) of acquisition optimization for a set of comparison methods, including EI as an additional method, on the *Alpine-2* function from the first experiment in our paper: ***EHIG: 6.9s, KG: 6.6s, EI: 0.5s, US: 0.3s***.

---

[3] https://en.wikipedia.org/wiki/Small_area_estimation
[4] https://earthobservatory.nasa.gov/features/NightLights

### B.1 Additional Experiment Results and Visualizations.

We show further experiment results for multi-level set estimation and sequence search (Figure 5), visualizations for multi-level set estimation (Figure 6), and an additional comparisons of classic BO acquisition functions on the initial top-$k$ optimization experiments (Figure 7).

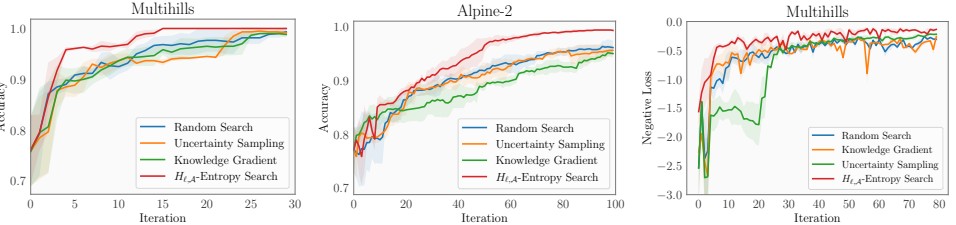

Figure 5: **Multi-level set estimation and sequence search.** *Left and center*: Plots of accuracy versus iteration for the task of multi-level set estimation (Equation (5), $m = 1$), where error bars represent one standard error. *Right*: Plot of negative loss versus iteration for the task of sequence search (Equation (6)), where error bars represent one standard error.

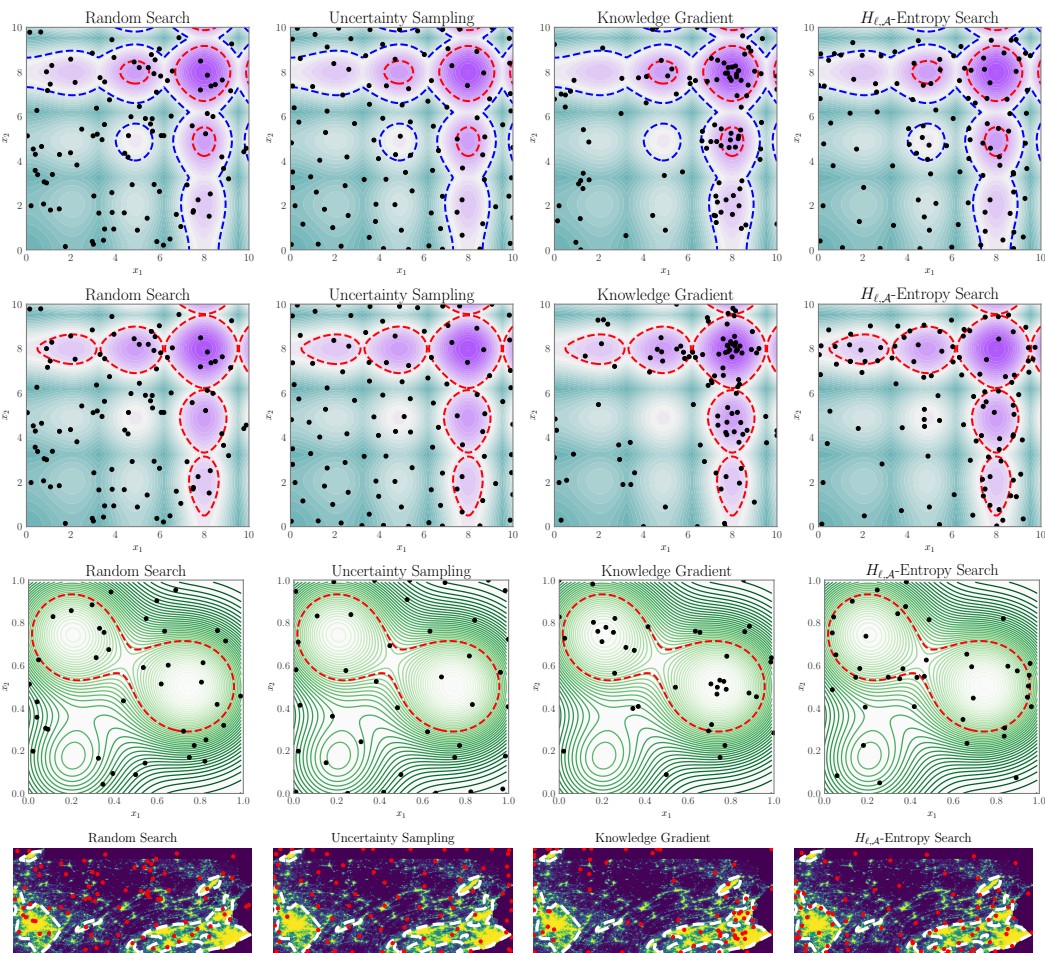

Figure 6: **Visualization results for multi-level set estimation.** Visualization of multi-level set estimation for Alpine-2, Multihills, and the Pennsylvania Night Light (PNL) functions. We show the ground-truth level set thresholds with red and blue dashed lines (for Alpine-2 and Multihills) and white dashed line (for the PNL function). The queries $\mathcal{D}_t$ taken by each method are shown with black dots (for Alpine-2 and Multihills) and red dots (for the PNL function). We observe that the queries taken by $H_{\ell,\mathcal{A}}$-Entropy Search focus on level set boundaries, yielding a fine-grained estimate near these boundary curves, while the other methods fail to do so.