# OpenReview forum: "Generalizing Bayesian Optimization with Decision-theoretic Entropies"
_NeurIPS.cc/2022/Conference — NeurIPS 2022 Accept_

### Official Review · Reviewer_YGMD · 2022-07-08

**Rating:** 6
**Confidence:** 3
**Soundness:** 3 good
**Presentation:** 4 excellent
**Contribution:** 3 good

**Summary:**

This work introduces a family of acquisition functions (AFs) for Bayesian Optimization (BO) based on the decision-theoretic entropies, $H_{l,A}$-$\textit{entropy}$, which is a generalized version of Shannon entropy. The AFs can be tailed to select queries that maximize the reduction of uncertainty in $H_{l,A}$-$\textit{entropy}$, which is defined as $\textit{expected}$ $H_{l,A}$-$\textit{information gain}$ (EHIG). The EHIG is a general form of AFs and can be reduced to information-based AFs or decision-theoretic AFs by choosing the parameters, $l$ and $A$. A framework is provided for the applications of EHIG on several categories of problems. Moreover, a gradient-based acquisition optimization method is proposed. Finally, evaluations of the method are made on examples datasets.

**Questions:**

1. Does the Random Search (RS) method optimize any function? In top row of Fig.2 the negative loss of RS slowly increases comparing to HES, but from the description of RS (line 305) the samples are drawn randomly from the full domain, therefore, it is hard to understand why the negative loss increases. Is there an explanation?

2. In Sequence Search task only the route of HES is shown and discussed. Bases on the left of the Fig. 4 at least the Uncertainty Sampling has comparable performance on the negative loss to HES. It would be better if all routes can be shown on the right of Fig. 4 for a thorough and clearer comparison.

3. Some of the plots are too small to see the details and axes, eg. bottom rows of Fig. 2 and Fig. 3.

**Limitations:**

There is no societal impact from this work. The authors should state and summarize the limitations clearly to the readers.


**Strengths And Weaknesses:**

Strengths:
1. This paper is well-organized and easy to read.
2. This article proposes a brand new general AF which unifies two branches, information-based and decision-theoretic AFs, for BO. Following the framework to carefully construct the the EHIG AF for each task and using the proposed gradient-based optimization method, the proposed $H_{l,A}$-$\textit{Entropy Search}$ (HES) procedure has advantage over other baseline methods.

Weaknesses:
1. The experiments are carried out on tasks which no AF has been developed in other work. However, to make this paper more significant, there should be examples that compare the result of the proposed HES method to other traditional BO method with AF on well-studied tasks. In this way the readers can have a better idea on the performance of HES.

---

> ### Author Response · Authors · 2022-08-02
> **Response to reviewer YGMD**
>
> Thank you for your helpful review! We appreciate the positive feedback, and address your comments and questions below.
>
> ### **Experiments on traditional optimization settings**
>
> Note that the main reason we strayed away from experiments on traditional settings (like vanilla optimization) is that, for the typical losses in this setting, our EHIG acquisition function is *equivalent* to existing acquisition functions, such as EI/KG/ES/PES. Therefore, for these traditional settings, we are not aiming to show improved performance of the EHIG over existing acquisition functions — as they should achieve roughly the same performance!
>
> We do, however, think that we make a contribution for these more-traditional settings, though not in terms of performance. Instead, for traditional optimization settings, our EHIG framework sheds light on when it is more suitable to choose one of the existing acquisition functions over the others (e.g. when to use EI vs KG vs ES vs PES), depending on the details of the optimization setting and the final error metric. This is because the EHIG sheds light on which acquisition function is optimal, depending on a problem-specific loss and action set to which the terminal action belongs, and thus gives guidance on which acquisition function to choose given the details of the optimization problem.
>
> While our HES method enjoys this unified perspective and provides new insights on the selection of existing acquisition functions according to different use cases, in our empirical study we thought it was more exciting to focus on the ability to easily adapt to new/customized optimization settings, which we think represents the next important step of applying (generalized) BO to broader applications. Thus we structured our experiments from this perspective.
>
> ### **Questions**
> 1. Although the random search (RS) baseline draws samples randomly from the domain, this does indeed optimize the negative loss, albeit quite slowly. For example, in Fig 2, these random samples do give some information about the top-$k$ points (with diversity) in the space, they just give far less information than the points chosen by HES (or the other baselines). If you consider an extreme case, where there are no limits on the sampling budget and we can draw an infinite number of random queries, we could then recover the black-box function up to any given accuracy — thus the curve will rise, but slowly compared to the other baselines.
>
> 2. Thanks for the suggestion here. In Figure 4, we primarily included the visual result to illustrate the sequence search task, and since the space was quite small, only included this single HES result so that the figure was not too crowded. However, we will definitely include a visualization of all methods in the appendix for a clearer comparison.
>
> 3. We will increase the font on both of these!
>
>
> **Thanks!** We hope that we have addressed each of your questions and comments. If there is anything else that we could do to increase your score, please let us know!

---

### Official Review · Reviewer_ryq5 · 2022-07-11

**Rating:** 6
**Confidence:** 4
**Soundness:** 3 good
**Presentation:** 3 good
**Contribution:** 3 good

**Summary:**

The authors derive a generic acquisition function design strategy (EHIG) based upon a user-specified loss functions and action set applicable to a variety of custom tasks involving Bayesian optimization, viz., Top-k optimization with diversity, Multi-level set estimation and Sequence search.
EHIG includes as special cases both information-based acquisition schemes such as entropy search (ES) and decision theoretic acquisition schemes such as knowledge gradient (KG) and expected improvement (EI).
Empirical results on multiple datasets indicate the superiority of EHIG on custom tasks considered.

**Questions:**

1. What distance function was considered for the diversity term in the experiments in Sec. 7 ? Was it Euclidean distance ? Did a parameter control the strength of the diversity penalty in eq. (4) ?

2. Did the authors only consider two levels (Multihills) and one level (Pennsylvania Night Light) for the multi-level set estimation experiments ?
Or were larger number of levels also considered ?


**Limitations:**

The authors discuss avenues for future work addressing implicit limitations in Sec 8.



**Strengths And Weaknesses:**

Strengths
1. The approach is clearly presented and well-motivated.

2. The proofs in the appendix seem to convincingly demonstrate how EHIG yields multiple well-known special cases such as ES, KG and EI.

3. Experimental validation is performed on a wide range of datasets.

Weaknesses

1. In the experimental comparison, only EHIG is tuned to the custom task and competing approaches are tuned to the conventional task of black box optimization (POM may be an exception in that it may be considered tuned for single level set estimation.)
So it appears to be a foregone conclusion that EHIG would outperform competitors.
It is unclear if any Bayesian Optimization approaches already exist for the custom tasks.

---

> ### Author Response · Authors · 2022-08-02
> **Response to reviewer ryq5**
>
> Thank you for your helpful review! We appreciate the positive feedback and aim to address each of your questions and comments below.
>
> ### **Baselines for custom tasks**
>
> > It is unclear if any Bayesian Optimization approaches already exist for the custom tasks.
>
> For our experiment results, we intentionally tried to focus on useful tasks where there does not already exist Bayesian optimization approaches that are specifically designed for these tasks (see our discussion on this in Section 1: Introduction, paragraph 4, and our note below the list of comparison methods in Section 7: Experiments).
>
> We feel these types of custom/novel settings are some of the best motivators for our EHIG framework, which can be customized to incorporate a problem-specific loss. Therefore, for these tasks in our experiments, we used—as far as we could determine—the best set of comparison acquisition functions that we could find as baselines for each task.
>
> ### **Question about distance function**
>
> For our experiments in Section 7, for the distance function in the diversity term of Equation (4), we indeed used the Euclidean distance. In this formulation, one could easily incorporate a parameter into the loss which controls the strength of the diversity penalty (and this seems like a nice idea to add!) — in general, we intend for the loss to be defined based on domain knowledge of a given problem at hand. We will add both details to the revised version of our paper.
>
> ### **Question about multi-level set estimation experiments**
>
> For our experiments in Section 7, we only went up to two levels. Our code, however, is written generally and can easily extend up to larger numbers of levels, though we didn’t focus on experiments on a higher number of levels in the paper.
>
> **Thank you**, and we hope that we have addressed each of your questions and concerns.

---

> > ### Comment · Reviewer_ryq5 · 2022-08-09
> > **Satisfactory response**
> >
> > The authors have adequately addressed my concern and questions.
> > I am glad they have also shown "Probability of Improvement" to be a special case of their approach in response to another reviewer.
> > My rating remains unchanged after considering the discussion between authors and reviewers thus far.

---

### Official Review · Reviewer_pn52 · 2022-07-15

**Rating:** 6
**Confidence:** 3
**Soundness:** 3 good
**Presentation:** 3 good
**Contribution:** 2 fair

**Summary:**

The paper generalises Shannon entropy-based acquisition functions (AF) to a broader class of uncertainty measures. Doing so allows the authors to frame several popular AFs as special cases of the proposed entropy. The proposed AF is also able to provide customised solutions for a number of modified versions of BO setting. The authors study efficient optimization of the AF under certain smoothness conditions. Experimental evaluation compares the proposed method with several baselines.

**Questions:**

A few questions in addition to the concerns raised in the previous section:

Line 115: How does the coin example help to motivate that this is a reasonable measure of uncertainty?

Experiments section: How many trials did the authors average results over for the experiments?

How realistic is Eq 7 in general?

Have the authors considered whether probability of improvement (the simplest BO AF one could imagine) can be obtained under the proposed framework?


**Limitations:**

-

**Strengths And Weaknesses:**

Post-rebuttal:
I would like to thank the authors for answering my questions. Most of my concerns are addressed, so I will update my score to 6.
__________

The paper is well written and appears to be technically sound. The idea of unifying the large zoo of BO acquisition functions under the same umbrella is neat (even though it was previously mentioned in the literature). It is particularly interesting to see both information-based and decision-theoretic AFs to be reached as special cases of proposed entropy.

The authors mention a diverse set of AFs for custom tasks. However, some of these applications seem somewhat artificially crafted. While the authors do provide a number of references, it’s not always clear on how such problems are derived from these references (e.g., line 221, reference 30 leads to a whole PhD thesis). Furthermore, with such limited description it's not always obvious why these tasks still require BO as an expensive black box optimization, and not some classic optimization techniques. It would be helpful if the authors could shed some more light on the motivation behind these custom tasks.

The experiments are one of the weaker points of the paper. While they do cover a diverse set of tasks, the set of methods is surprisingly small and lacks some “must-have” classic BO baselines, such as PI or GP-UCB. Even more surprisingly, EI, which is mentioned in the main paper as a special case of proposed entropy, is missing in the experimental section.

---

> ### Author Response · Authors · 2022-08-02
> **Response to reviewer pn52**
>
> Thank you for your helpful review. We appreciate the positive feedback on our paper (“well written”, “technically sound”, "particularly interesting", “unifying the large zoo of BO acquisition functions under the same umbrella is neat”), and aim to address each of your comments and questions below in order to improve the quality of our submission.
>
> ### **Motivation behind custom tasks:**
>
> To provide some additional motivation for the custom tasks in our experiments, we wanted to describe a few more concrete instances where these tasks are useful in practice.
>
> We see the task of *top-k optimization with diversity* whenever we have an expensive black-box function, and want to estimate multiple optimal designs (or locations, etc) and don’t want redundancy in the optimal designs. One example motivation is in tasks such as active monitoring of pollution [1], if a user wishes to efficiently estimate a set of locations that have the highest levels of pollution, in order to allocate sensors for aid/regulation or resources for cleanup. Another example is in the space of materials design, such as in computational catalyst screening [2], where the goal is to perform a sequence of expensive simulations in order to efficiently determine the top-k catalysts with highest simulated adsorption energies, as a recommended set for follow-up experiments.
>
> Another recent application that we are familiar with from the work of our colleagues, which is currently underway, is in the space of materials/mixture characterization. Here, the goal is to guide temperature and pressure controls in Small Angle X-ray Scattering measurements, in order to efficiently characterize properties of a class of supercritical fluids (SCFs) [3]. Notably, in this application, a set of two peaks in the measurement space must be found in order to characterize the SCF properties of interest. This is precisely a top-two optimization problem as described in our paper, and a top-k-with-diversity acquisition function is currently being developed for this task in practice.
>
> For the *sequence search* task, we also find concrete applications as well. For example, applications of this appear in materials design, in the task of synthesizing a library of nanoparticle sizes [4] — i.e. where the goal is to find a set of inputs that yield a set of nanoparticles of different pre-defined sizes. Finally, *multi-level-set estimation* is useful any time one needs to estimate more than two partitions of a design space. This is useful in various applications, such as when estimating phase boundaries for materials design [5], or when health policy makers must estimate multiple disease prevalence level sets (i.e. regions where COVID prevalence exceeds 1%, 2%, etc.) for graded reopening decisions [6, 7].
>
> Note that our Bayesian-model-based methods have particular benefits over classic optimization techniques in cases where we can only get zeroth order information via function evaluations (i.e. no gradients), and where the function is particularly expensive — and thus we need to be as sample (iteration) efficient as possible. These Bayesian techniques allow us to leverage our probabilistic surrogate model to intelligently choose a sequence of function queries for increased sample efficiency.
>
> We will include some of these concrete applications, more specific citations, and above discussion in the updated version of our paper.
>
> [1] S. P. Hellan, C. G. Lucas, N. H. Goddard. Bayesian Optimisation for Active Monitoring of Air Pollution. In 36th AAAI Conference on Artificial Intelligence, 2021.
>
> [2] K. Tran, W. Neiswanger, K. Broderick, E. Xing, J. Schneider, Z. Ulissi. Computational catalyst discovery: Active classification through myopic multiscale sampling. The Journal of Chemical Physics, 2021. https://doi.org/10.1063/5.0044989
>
> [3] K. Nishikawa, I. Tanaka. Correlation lengths and density fluctuations in supercritical states of carbon dioxide. Chemical physics letters, 1995.
>
> [4] A. Fong, L. Pellouchoud, M. Davidson, R. Walroth, C. Church, E. Tcareva, L. Wu, K. Peterson, B. Meredig, C. Tassone. Utilization of machine learning to accelerate colloidal synthesis and discovery. J. Chem. Phys, 2021. https://doi.org/10.1063/5.0047385
>
> [5] D. Pradhan, S. Kumari, E. Strelcov, D. Pradhan, R. Katiyar, S. Kalinin, N. Laanait, R. Vasudevan. Reconstructing phase diagrams from local measurements via Gaussian processes: mapping the temperature-composition space to confidence. Nature Computational Materials. 2018.
>
> [6] E. Oh, A. Mikytuck, V. Lancaster, J. Goldstein, S. Keller. Design and Estimation for the Population Prevalence of Infectious Diseases. medRxiv, 2021.
>
> [7] C. Yiannoutsos, P. Halverson, N. Menachemi. Bayesian estimation of SARS-CoV-2 prevalence in Indiana by random testing. Proceedings of the National Academy of Sciences, 2021.
>
> **(Response continued in comment below)**

---

> > ### Author Response · Authors · 2022-08-02
> > **Response to reviewer pn52 - Continued**
> >
> > **(Continued from comment above)**
> >
> > ### **Lacks some classic BO baselines, such as PI or GP-UCB:**
> >
> > We are very happy to include these standard BO baselines (such as PI and GP-UCB) and have added them to the revised paper for a few initial experiments (Appendix B.1, Figure 7).
> >
> > When writing the paper, we originally thought to disclude these baselines — the reason was because we chose to focus our experiments on tasks such as top-$k$ optimization and other custom settings (rather than vanilla optimization), and since these classic acquisition functions like PI/GP-UCB are not designed for these custom settings we thought it didn’t make sense to include more than one “vanilla optimization” baseline (for which we chose knowledge gradient).
> >
> > That being said, we are happy to include PI/GP-UCB as suggested! We’ve added plots that show the comparison among these acquisition functions in the first experiment (Figure 7) so far, and will add them to the further experiments as well if it is desired.
> >
> > ### **EI in experiments**
> >
> > Similar to the discussion above, based on our experiments on non-vanilla optimization settings, we wanted to focus on losses under our HES framework that were tailored to these tasks. This is in contrast with EI, which we prove (in Section 4) is an instance of our framework that is well-suited for vanilla optimization.
> >
> > However, we are happy to include EI experimental comparisons if these are desired — for a start, we’ve implemented/run this baseline, and added the results to the comparison plots for the initial experiments in our revised paper (Appendix B.1, Figure 7), and will add them to the further experiments if desired.
> >
> > ### **Probability of improvement as an example of proposed framework**
> >
> > Thanks for bringing up this discussion. When we originally worked on our submission, we had trouble coming up with a good way to fit the probability of improvement (PI) acquisition function within our EHIG framework, and did not pursue it further. However, we made another attempt based on your suggestion and believe that we’ve found a way to incorporate this acquisition function. We’ve added a Theorem and Proof to Appendix Section A.4 that shows how (PI) is a special case of our EHIG framework under a particular loss and action set—specifically that EHIG can be made equivalent to PI up to an additive constant. Note that this proof is similar to our proof for the expected improvement (EI) acquisition function.
> >
> > ### **Coin flip example to motivate a reasonable measure of uncertainty**
> >
> > The coin flip example was simply intended to help explain the definition of a concave uncertainty measure, and also give intuition of why a concavity is a desirable property of an uncertainty measure, as it is a property of both $H_{\ell, \mathcal{A}}$-entropy as well as Shannon entropy. Specifically, this concavity property means that the average of uncertainties of two distributions should be less than the uncertainty of the average (mixture) distribution. In this coin flip example (where we “have two distributions p1 and p2, and flip a coin to sample from p1 or p2,”), this concavity property is equivalent to saying that we should have less uncertainty about the final sample if we are allowed to observe the outcome of the coin flip than if we are not allowed to observe it — which makes intuitive sense as a property that we want!
> >
> > **Thanks!** We hope that we have addressed your concerns — If there is anything else that we could do to increase your score, please let us know!

---

### Meta-Review · Area_Chair_4xxr · 2022-08-28

**Recommendation:** Accept
**Confidence:** Certain

**Metareview:**

The paper proposed a novel acquisition function for BO, based on a generalization of Shannon entropy that enables one to incorporate problem-specific loss functions corresponding to a downstream task. The authors show that the proposed acquisition criterion generalizes a number of well-known BO acquisition functions, including EI/KG/ES/PES. A detailed training procedure for optimizing the acquisition function was discussed in the paper, and experimental results show that the proposed acquisition function with the optimization procedure performs well over a diverse set of tasks.

All reviewers agree that this paper is well written, and the idea of unifying a collection of “classical” BO acquisition functions is interesting. There were a few concerns about the sufficiency/significance of the experiments, mainly due to the (lack of) baselines considered in the tasks. The authors clarified the concerns by including preliminary runs of several new experiments, and highlighting that the proposed approaches were targeting novel tasks that went beyond the vanilla optimization tasks. There were no other critical concerns in the reviews. The authors are strongly encouraged to address the questions raised in the reviews when preparing a revision of this paper.


**Award:**

No

---

### Decision · Program_Chairs · 2022-09-14

Accept